# Design Study for an Airborne N$_2$O Lidar

Christoph Kiemle, Andreas Fix, Christian Fruck, Gerhard Ehret, Martin Wirth

Deutsches Zentrum für Luft- und Raumfahrt (DLR), Institut für Physik der Atmosphäre, D-82234 Oberpfaffenhofen, Germany

*Correspondence to*: Christoph Kiemle (Christoph.Kiemle@dlr.de)

**Abstract.** Nitrous oxide (N$_2$O) is the third most important greenhouse gas modified by human activities after carbon dioxide and methane. This study examines the feasibility of airborne differential absorption lidar to measure N$_2$O concentration enhancements over agricultural, fossil fuel combustion, industrial and biomass burning sources. The mid-infrared spectral region, where suitably strong N$_2$O absorption lines exist, challenges passive remote sensing by means of spectroscopy due to

both low solar radiation and thermal emission. Lidar remote sensing is principally possible thanks to the laser as independent radiation source, but has not yet been realized due to technological challenges. Mid-infrared N$_2$O absorption bands suitable for remote sensing are investigated. Simulations show that a spectral trough position between two strong N$_2$O lines in the 4.5 µm band is the favored option. A second option exists in the 3.9 µm band at the cost of higher laser frequency stability constraints and less measurement sensitivity. Both options fulfill the N$_2$O measurement requirements for agricultural areal or

point source emission quantification (0.5 % measurement precision, 500 m spatial resolution) with technically realizable and affordable transmitter (100 mW average laser power) and receiver (20 cm telescope) characteristics for integrated-path differential absorption lidar that measures the column concentration beneath the aircraft. The development of an airborne N$_2$O lidar is feasible yet would benefit from progress in infrared laser transmitter and low-noise detection technology. It will also serve as a precursor to space versions which are still out of reach due to the lack of space technology.

## 1 Introduction

The average concentration of nitrous oxide (N$_2$O) amounts to only 337 ppb yet its global warming potential is nearly 300 times that of CO$_2$ on a 100-year span (Arias et al., 2021). This makes N$_2$O the third most important greenhouse gas contributing to human-induced global warming after carbon dioxide and methane. The major anthropogenic source is nitrogen fertilization on arable land. Further N$_2$O sources are processes in the chemical industry and combustion processes.

According to current knowledge, anthropogenic sources contribute ~36 % to total global N$_2$O emissions (Tian et al., 2024). Emissions from natural soils and oceans constitute the major natural sources. Agricultural N$_2$O emissions are increasing due to interactions between nitrogen inputs and global warming, constituting an emerging positive N$_2$O-climate feedback. The recent increase in global N$_2$O emissions exceeds even the most pessimistic emission trend scenarios developed by the IPCC, underscoring the urgency to mitigate N$_2$O emissions (Tian et al., 2024). Estimating N$_2$O emissions from agriculture is

inherently complex and comes with a high degree of uncertainty, due to variability in weather and soil characteristics, in

agricultural management options and in the interaction of field management with environmental variables (Eckl et al., 2021). Moreover, $N_2O$ measurements are sparse. Consequently, more comprehensive $N_2O$ concentration measurements are needed, particularly by means of remote sensing. Recently, the World Meteorological Organization has launched the Global Greenhouse Gas Watch (G3W) initiative to endorse, among others, this need (WMO, 2024).

The mid-infrared (mid-IR) spectral region, where suitably strong $N_2O$ absorption lines exist, challenges passive remote sensing by means of spectroscopy due to both low solar radiation and thermal emission from Earth's surface (e.g., Ricaud et al., 2021; Vandenbussche et al., 2022). Lidar remote sensing is principally possible thanks to the laser as independent radiation source, but has not yet been realized due to technological challenges. While mid-IR lidars are employed for ground-based pollution detection (e.g., Robinson et al., 2011; Gong et al., 2020), to our knowledge, neither an airborne $N_2O$

lidar has been realized yet, nor has passive remote sensing by means of spectroscopy been used to measure $N_2O$ concentrations in the lower troposphere.

Airborne $N_2O$ lidar remote sensing has the potential to combine the advantages of high measurement accuracy, large-area coverage and dawn/dusk or nighttime measurement capability to study diurnal concentration variations. Initial studies have shown that Integrated-Path Differential-Absorption (IPDA) lidar from an airborne or even a satellite platform has the

45 potential to measure $N_2O$ with high precision and low bias (Ehret and Kiemle, 2005; Ehret et al., 2008). It uses the Earth surface backscatter signal at an "online" laser wavelength tuned to a $N_2O$ absorption line to obtain column concentrations of $N_2O$ (Ehret et al., 2008; Amediek et al., 2017). A parallel reference measurement at the non-absorbed "offline" wavelength avoids biases generated by albedo variations or aerosol layers within the light path. In comparison to conventional lidar using backscatter from atmospheric molecules and aerosol, IPDA lidar yields high signal to noise ratio at comparatively low

instrument size since the surface backscatter is about 100 times stronger than the atmospheric backscatter in terms of laser energy per range gate. Still, accurate ranging by means of short laser pulses is important for precise measurements of the individual column length. First airborne systems for $CO_2$ and $CH_4$ have demonstrated high measurement accuracy and the capability to measure in broken cloud environments (Amediek et al., 2017; Sun et al., 2021; Barton-Grimley et al., 2022; Mao et al., 2024).

The objective to quantify agricultural areal or point source emissions requires obtaining $N_2O$ column concentration gradients along the flight track between background levels outside the emission regions and the $N_2O$ source regions, cultivated soils or exhaust plumes from, e.g., fertilizer production sites. The airborne lidar should point downward from a flight altitude of about 5 km, well above the boundary layer in which $N_2O$ surface emissions disperse vertically by turbulence. Gradients over agricultural regions measured by airborne in-situ instruments (Eckl et al., 2021; Waldmann et al., 2024) suggest that the

maximum uncertainty of the $N_2O$ column measurement should be 0.5 % and that an along-track horizontal measurement resolution of 500 m is sufficient. Consequently, the measurement instability due to instrumental drifts or changing biases should remain below 0.5 %. Experience from airborne lidar campaigns shows that long-term stability can be controlled by executing repeated flight legs over the same tracks, and over background concentrations in case those can be assumed

constant. To detect smaller but denser N$_2$O emission plumes from industrial production sites the horizontal resolution can be improved at the cost of precision. Lidar allows such tradeoffs to adapt to the measurement objectives.

This study first investigates the N$_2$O spectroscopy to find suitable absorption lines. The chosen wavelength has consequences for the surface reflectance, the atmospheric absorption, the solar and thermal background radiation, as well as transmitter and detector options. All relevant environment, instrument and spectroscopic constraints are implemented into a lidar simulation model to design the instrument in order to meet the above measurement requirements. Finally, although beyond the scope of this study, concepts for suitable lidar transmitter and detector technologies are briefly discussed.

## 2 N$_2$O Spectroscopy

Up-to-date spectroscopic data retrieved with the HITRAN (high resolution transmission molecular absorption database) Application Programming Interface (HAPI) are used to find suitable absorption lines in the four major rotational-vibrational N$_2$O bands located at 2.9, 3.9, 4.5 and 7.8 µm (Nemtchinov et al., 2004; Loos et al., 2015; Kochanov et al., 2016; Gordon et al., 2022). Molecular absorption cross sections are calculated line-by-line with a resolution of 0.001 cm$^{-1}$ with the Hartmann-Tran scheme (Ngo et al., 2013) using standard atmosphere profiles of pressure, temperature and trace gas concentrations within the lowest 5 km, below the foreseen flight altitude. The vertically integrated product of the absorption cross section σ and the trace gas molecule number density n, both varying with altitude z between the surface sfc and the flight height flh, is the optical depth od:

$$od_{gas}(\lambda) = \int_{sfc}^{flh} \sigma_{gas}(z,\lambda) \cdot n_{gas}(z) \, dz \tag{1}$$

It is related to atmospheric transmission and represents the spectroscopic determining parameter for IPDA lidar column measurements. The IPDA technique is well described in e.g. Ehret et al. (2008) and Amediek et al. (2017). Criteria for line selection are (a) trace gas molecule number density, (b) appropriate line strength, related to the optimal optical depth, (c) low temperature sensitivity and (d) minimal influence by other gases. The optimal line strength or optical depth is a compromise between too weak absorption and saturation. The optimal optical depth typically lies between 0.5 and 1.2 and depends on column height and instrument noise (Ehret et al., 2008). Temperature sensitivity is determined using an atmospheric temperature profile shifted by 1 K and evaluating the difference in optical depth between the reference and the temperature-shifted optical depth.

The N$_2$O bands at 2.9 and 3.9 µm contain absorption lines of comparable strength, yet the entire 2.9 µm N$_2$O band is dominated by water vapor absorption lines whose wings are without exception stronger than the N$_2$O lines within the lowest 5 km, hence the 2.9 µm band is not shown here. The 3.9 µm N$_2$O band is illustrated in Fig. 1 showing the one-way column optical depth of the lowermost 5 km as function of vacuum wavenumber and wavelength. The 3.9 µm band is composed of

relatively weak lines suitable for satellite lidar (Ehret and Kiemle, 2005; Ehret et al., 2008) yet suboptimal for lower tropospheric concentration measurements. One of the strongest absorption lines, situated at 2576.54 cm$^{-1}$, is selected here (red line), characterized by both low influence of other absorbing gases and low temperature sensitivity. The offline reference wavenumber is set to a neighboring minimum absorption and temperature sensitivity position.

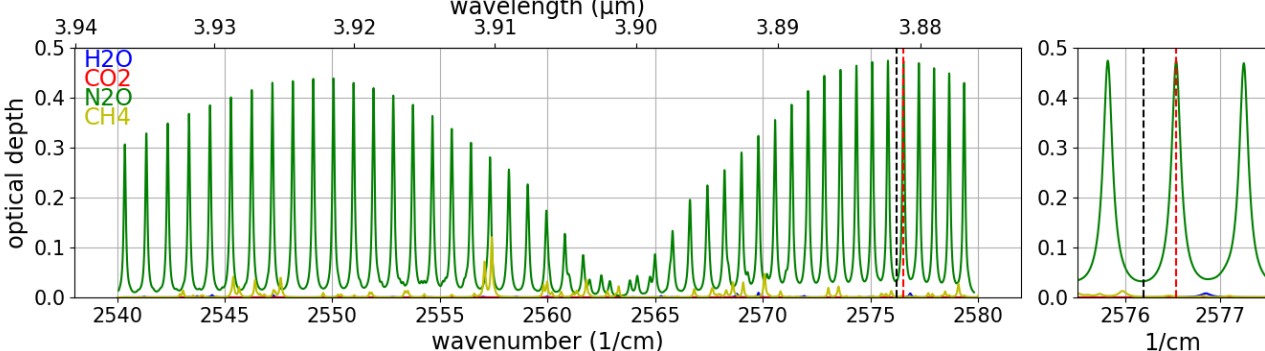

**Figure 1:** *Optical depth spectrum of trace gases in the 3.9 μm N$_2$O band in a standard atmosphere vertical column covering the lowest 5 km. The selected online (offline) position is highlighted in red (black) and within a close-up (right). Absorption lines of N$_2$O (green), water (blue), CO$_2$ (red) and CH$_4$ (yellow) need to be considered in this spectral region.*

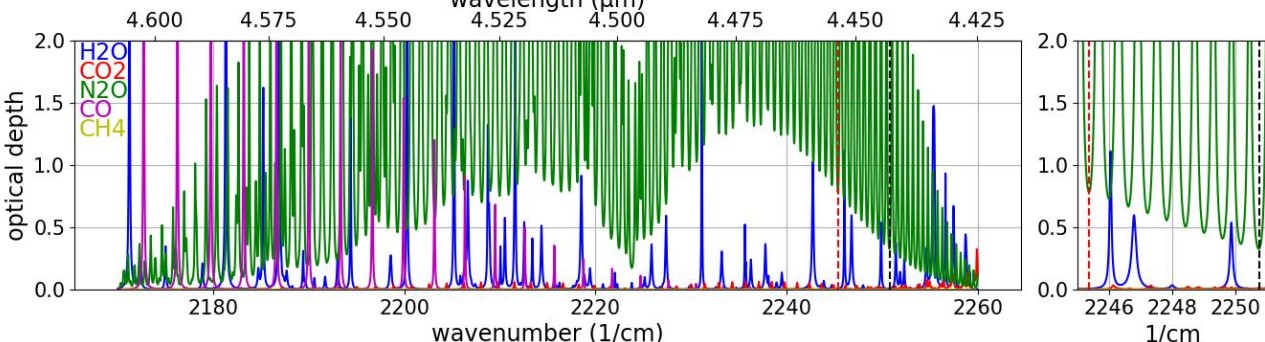

**Figure 2:** *Optical depth spectrum of trace gases in the 4.5 μm N$_2$O band in a standard atmosphere vertical column covering the lowest 5 km. The selected online (offline) position is highlighted in red (black) and within a close-up (right). Absorption lines of CO have to be considered in addition (magenta).*

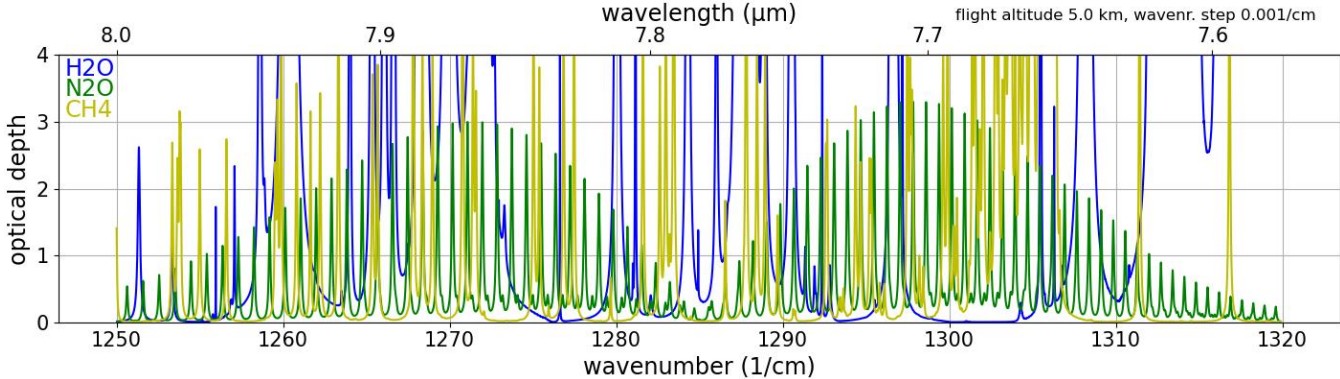

**Figure 3:** *Optical depth spectrum of trace gases in the 7.8 μm N₂O band, as is Figures 1 and 2. The atmospheric window ends at 1310 cm⁻¹ where a large water vapor absorption band begins.*

The 4.5 μm band contains much stronger $N_2O$ lines (Fig. 2). $N_2O$ lines with suitable strength exist at both edges of the band, around 2180 and 2255 cm⁻¹, as well as in the center of the band. However, all of those reveal inappropriate due to overlapping water lines and high temperature sensitivity in the center of the band. Finally, Figure 3 illustrates the 7.8 μm band which is so densely populated with strong water and methane lines that no suitable $N_2O$ line candidate is found. Water vapor is difficult to correct due to its high variability in the lower troposphere. Consequently, we restrict this study to the 3.9

and 4.5 μm $N_2O$ bands. In the 4.5 μm band, trough positions at minimum optical depth in between two strong lines can be selected for both on- and offline, as used for $CH_4$ lidar and foreseen for MERLIN, the Remote Sensing Methane Lidar Mission (Kiemle et al., 2011; Amediek et al., 2017). A trough position has two advantages. First, it relaxes the laser frequency stability requirement due to a relatively flat optical depth in the center of the trough. More quantitatively, the derivative of optical depth with respect to wavenumber around the minimum of the trough is a factor fifty to hundred lower

than outside the trough in the steep flank of a line (Kiemle et al., 2011). Second, the measurement sensitivity at low altitudes, i.e. near the surface where the $N_2O$ sources are located, is improved thanks to pressure broadening of both lines surrounding the trough (Ehret et al., 2008). Figure 4 illustrates the differences in sensitivity between line center, wing and trough (i.e. far wing) positions. Within the lowest 1 km a spectral position in the center of the trough is found to increase the near-surface sensitivity to $N_2O$ by about a factor of 1.5 (= 1.4 % / 0.92 %) in comparison to a line center position.

Therefore, in the 4.5 μm band we select a trough position at the high-wavenumber side of the band at 2245.35 cm⁻¹ because it is less influenced by other gases than the low-wavenumber side (Fig. 2). The closest possible offline position is at 2250.75 cm⁻¹. The 10.7 nm distance from the online position may require separate lasers for the generation of the on- and offline wavelengths and may lead to uncertainties estimated to < 1 % if surface albedo or aerosol extinction are wavelength

dependent (Amediek et al., 2009). Table 1 summarizes the spectroscopic characteristics of the candidate lines in both bands. The temperature sensitivity is sufficiently low, especially at the online positions. The additional optical depth by line wings

of other gases is insignificant at 3.9 µm and ~0.01 at 4.5 µm. In the event of a 50 % concentration change possible for $H_2O$ yet very unlikely for $CO_2$, the impact on the total optical depth at 4.5 µm would be 0.005/0.79 ≈ 0.6 % for the online and 0.005/0.34 ≈ 1.5 % for the offline, which we consider uncritical. Note that our line selection is provisional as new spectroscopic data may lead to better options.

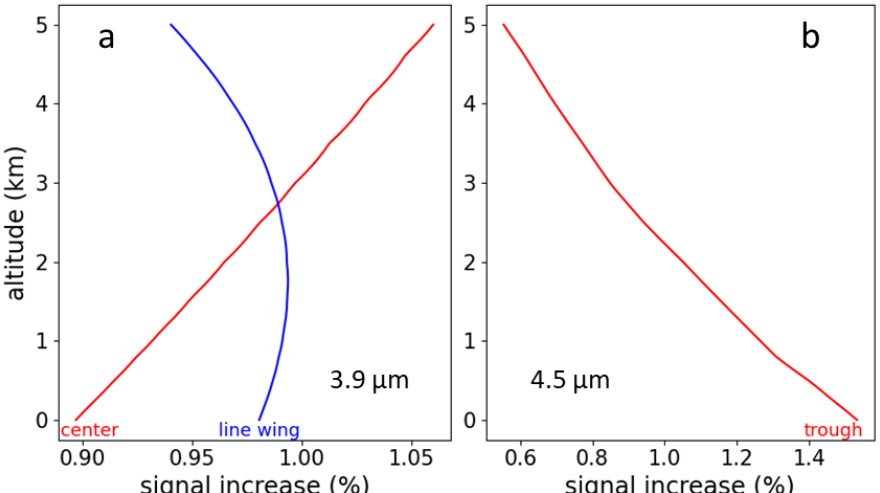

**Figure 4:** *$N_2O$ measurement sensitivity expressed as online optical depth (signal) increase under the assumption of a hypothetical 50 % $N_2O$ concentration increase within a 100 m thick layer, as function of the altitude of this layer. (a) 3.9 µm line center (red) position of Fig. 1 in comparison with an online line wing (blue) position at 2576.57 $cm^{-1}$ (very close to the line center; not further discussed). (b) 4.5 µm online trough position of Fig. 2.*

| Line selection | wavenumber $cm^{-1}$ | $N_2O$ opt. depth (0-5 km) | temperature sensitivity | opt. depth of other gases (0-5 km) |
|---|---|---|---|---|
| 3.9 µm online, line center | 2576.54 | 0.47 | - 0.01 % / K | 0.002 ($CH_4$) |
| 3.9 µm offline | 2576.20 | 0.03 | 0.43 % / K | 0.001 ($CH_4$) |
| 4.5 µm online, trough | 2245.35 | 0.78 | 0.06 % / K | 0.008 ($H_2O$) |
| 4.5 µm offline | 2250.75 | 0.33 | - 0.31 % / K | 0.008 ($H_2O$), 0.010 ($CO_2$) |

**Table 1:** *Selected online and offline spectral positions in the 3.9 and 4.5 µm $N_2O$ absorption bands with corresponding $N_2O$ optical depth for a vertical standard atmosphere column in the lowest 5 km, relative $N_2O$ optical depth change per Kelvin temperature change, and optical depth of other trace gases.*

## 3 Mid-IR Surface Albedo, Aerosol Influence and Background Radiation

The albedo is key for IPDA lidar which relies on surface backscatter intensity. It is generally low in the IR, compared to the near-IR and visible spectral ranges. Ehret and Kiemle (2005) and Ehret et al. (2008) used a value of 0.02 (2 %) for land surfaces. The Ecosystem Spaceborne Thermal Radiometer Experiment on Space Station (ECOSTRESS) spectral library contains a limited set of mid-IR reflectance data (Baldridge et al., 2009; Meerdink et al., 2019). Table 2 lists their agriculturally relevant values. We keep the small value of 2 % from our initial studies for the present simulation which can be considered a safe worst case. Over agricultural soils, extinction by rural aerosol within the boundary layer is expected. Based on IR lidar measurements by Vaughan et al. (1995 and 1998) and scaled to 3.9 and 4.5 µm using an Angström exponent of 1, we assume a worst-case maximum aerosol optical depth of 0.2 across the lowest 5 km. The Rayleigh optical depth due to air molecular extinction amounts to $\sim 2 \cdot 10^{-5}$ at around 4 µm in the lowest 5 km and is therefore negligible.

Photons from solar scattered and thermal emitted radiation cause noisy background signals in the detector. The mid-IR radiation emitted from earth's surface is calculated with Planck's law (Stull, 2017) assuming a blackbody with albedo zero, at 288 K temperature and without atmosphere. According to Kirchhoff's law this represents the worst-case maximum thermal emission, also because the atmosphere is mostly colder than earth's surface. The solar radiation is calculated with Planck's law assuming an albedo of 0.4, a sun in the zenith and no atmosphere, which also represents a maximum-radiation worst case. Table 2 summarizes the environmental boundary conditions for the lidar simulation. The low total radiation, close to the spectral minimum of the sum of thermal emitted and solar scattered radiation represents a challenge for passive remote sensing in the mid-IR.

| N₂O band | 3.9 µm | 4.5 µm | unit | remarks |
|---|---|---|---|---|
| Albedo | 0.03-0.04 | **0.02**-0.03 | - | over grain or grass |
| | 0.23-0.32 | 0.13-0.35 | | over soil |
| Aerosol optical depth, $od_a$ | 0.2 | 0.2 | - | 0 - 5 km |
| Earth emission | 0.4 | 1.0 | W / (m² µm sr) | albedo 0.0 and 288 K assumed |
| Solar backscatter | 1.2 | 0.8 | W / (m² µm sr) | albedo 0.4 assumed |
| Total background radiation | 1.6 | 1.8 | W / (m² µm sr) | sum of thermal emitted and solar scattered radiation |
| N₂O $od_{on} - od_{off}$ | 0.44 | 0.45 | - | see Eqs. (1) and (3) |

**Table 2:** *Mid-IR albedo, aerosol optical depth and background radiation used for the N₂O lidar simulation. The minimum albedo of 0.02 (bold) is selected as a worst case for the simulation. Soil has about ten times higher albedo than grain or grass. The differential absorption optical depth is about equal in both N₂O bands.*

## 4 N₂O Lidar Simulation

The airborne lidar performance is assessed by implementing the measurement requirements together with environmental, instrumental and spectroscopic constraints into a lidar noise propagation model developed for earlier studies (Kiemle et al., 2011 and 2017). Atmospheric transmission and surface scattering are evaluated to compute the backscattered signal power as function of emitted laser power $P_L$, surface reflectance $\rho$ (= albedo / $\pi$), receiver optical efficiency $\eta$, telescope area A, and range R (= 5 km) for the on- and offline wavelengths in the IPDA lidar equation:

$$P_{on,off} = P_{L,on,off} \, \varrho \, \eta \, A \, R^{-2} \, exp^{-2 \, (od_a + od_{on,off})} \tag{2}$$

According to the Beer-Lambert law the exponential term represents the atmospheric transmission along the vertical path, lowered by the aerosol optical depth $od_a$ and the N₂O on- and offline optical depths. Solving Eq. (2) for the respective on-

195 and offline optical depths, and assuming constant surface albedo, optical efficiency, and aerosol optical depth for both on- and offline wavelengths, we obtain the differential absorption optical depth DAOD from subtracting the offline solution of Eq. (2) from the online solution:

$$DAOD = od_{on} - od_{off} = \frac{1}{2} ln \left( \frac{E_{L\_off}/E_{ref\_off}}{E_{L\_on}/E_{ref\_on}} \right), \tag{3}$$

where $E_{L\_on,off}$ are the received on-/offline laser pulse energies, and $E_{ref\_on,off}$ are energy reference measurements accounting for variations of the emitted pulse power $P_L$. After Eq. (1) the DAOD is proportional to the N₂O column concentration weighted with the absorption cross section. Table 2 lists the DAOD values expected in a reference atmosphere.

Table 3 lists the major instrument parameters. The larger the average laser power, surface reflectance (albedo), and telescope size, the stronger the received signal power and consequently the signal-to-noise ratio (SNR). Further parameters influencing the SNR are the flight altitude, the horizontal resolution, the laser pulse repetition rate, the receiver field-of-view and several detector parameters. Parametric analyses allow to finetune the instrument with the aim to optimize efficiency or flexibility. Parameters can depend on each other, such as the aircraft velocity, the horizontal resolution, the repetition rate, and the

number of averaged pulses. Likewise, for the laser, repetition rate, pulse energy and average laser power are related. Finally, the telescope field-of-view is related to its diameter, its focal ratio, and the detector size. Equations implement all relevant dependencies in the model. For more details we refer to our former studies (Ehret et al., 2008; Kiemle et al., 2011 and 2017).

| Simulation parameter | value | reference |
|---|---|---|
| **Requirements: flight altitude** | **5 km** | Eq. (2): R |
| **horizontal along-track resolution** | **500 m** | |
| aircraft velocity | 150 m/s | |
| **Laser**: **average IR online and offline power** | **100 mW** | Eq. (2): $P_{L,on,off}$ |
| pulse energy | 0.5 mJ | |
| pulse energy reference measurement precision | 1 % | Eq. (3): $E_{ref\_on,off}$; Ehret et al. (2008) |
| double pulse (online, offline) repetition rate | 100 Hz | |
| number of averaged double pulses | 333 | |
| pulse length | 15 ns | Ehret et al. (2008) |
| beam divergence, full angle | 0.6 mrad | |
| spectral line width | 30 MHz | Ehret et al. (2008) |
| frequency stability at 3.9 μm | ~ 1 MHz | Ehret et al. (2008) |
| frequency stability at 4.5 μm | ~ 100 MHz | Kiemle et al. (2011) |
| **Receiver**: **telescope area** | **0.03 m$^2$** | Eq. (2): A (20 cm diam.) |
| telescope focal ratio, 1/f | 1.25 | |
| **optical efficiency** | **0.65** | Eq. (2): η; Kiemle et al. (2011) |
| optical narrow-band filter | 1 nm | Ehret et al. (2008) |
| field-of-view, full-angle | 0.8 mrad | |
| footprint size at surface | 4 m | |
| **Detector: MCT APD, NEP** | **0.1 pW/Hz$^{0.5}$** | e.g., Sun et al. (2017) |
| diameter | 200 μm | e.g., Martyniuk et al. (2023) |
| bandwidth | 3 MHz | Ehret et al. (2008) |
| **Surface albedo** | **0.02** | Eq. (2): ρπ; Ehret et al. (2008) |

**Table 3:** *$N_2O$ lidar instrument parameters assumed for the simulation, partly adopted from earlier studies and regarded as achievable. Parameters of first importance to the performance are in bold.*

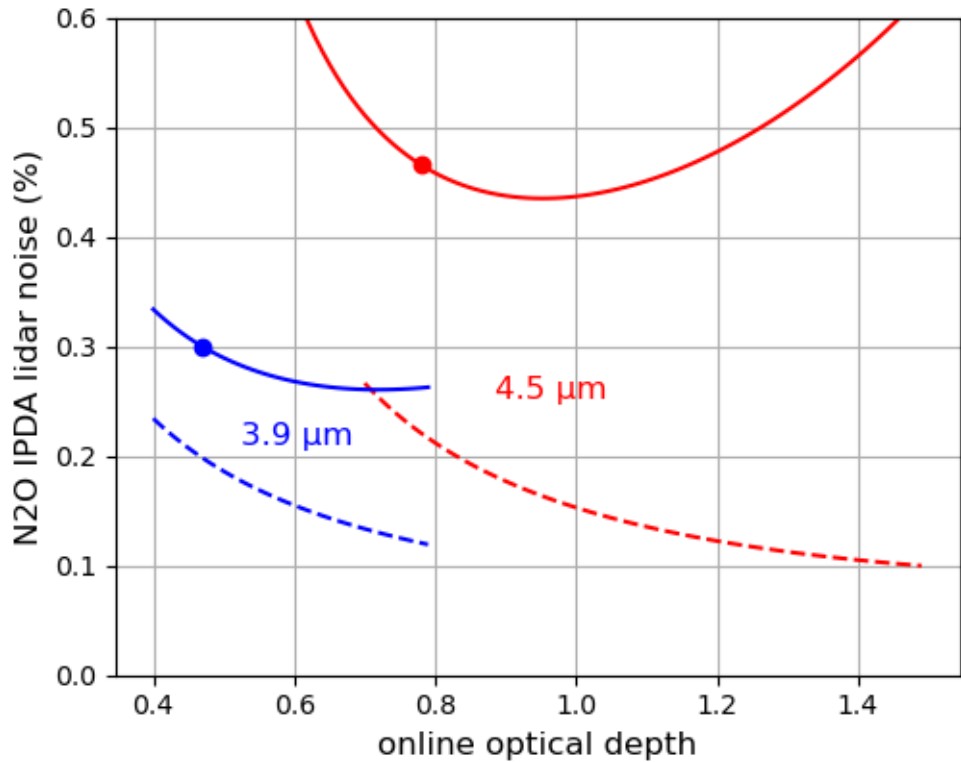

*Figure 5: Simulated N₂O lidar measurement precision at 3.9 (blue) and 4.5 µm (red) as function of online optical depth under the conditions of Tables 2 and 3. The dots indicate the selections of Table 1 and the dashed lines represent an ideal noise-free detector for comparison. The absence of strong lines in the 3.9 µm band limits the 3.9 µm online optical depth range.*

The simulation is run with the environment conditions of Table 2 and the instrument parameters of Table 3. All noise contributors, Poisson noise from laser photons and background radiation, detector noise, laser speckle noise within the field of view, and energy reference measurement noise are considered. Speckle noise can be more significant for laser measurements in the mid-IR since the speckle cell sizes are larger and there are fewer "speckles" (regions of constructive interference) on the detector surface. This effect is compensated by a larger telescope field-of-view and laser beam

divergence, compared to near-IR applications. As all noise sources are basically uncorrelated, error propagation on the basis of Eqs. (2) and (3) provides the overall one-sigma precision of the N₂O measurements as function of the prescribed online optical depth for both N₂O bands (Fig. 5). The offline optical depth is kept at a constant level of 0.03 (0.33) at 3.9 (4.5) µm. At 3.9 µm this level corresponds to a representative minimum optical depth within the band (Fig. 1), while at 4.5 µm the

offline from Table 1 may serve several neighboring online trough options (Fig. 2). The dashed lines show the performance of an ideal noise-free detector for comparison. While the detector noise is assumed zero, all other noise contributors remain: Poisson / shot noise, speckle noise, and energy measurement noise. Their minima are right-shifted because the optimum

online optical depth is larger under low-noise conditions (Ehret et al., 2008). The curves are flatter since a low-noise instrument is more tolerant with respect to suboptimal spectroscopic settings.

Assuming, as a provisional proxy, a mercury cadmium telluride (MCT) avalanche photodiode (APD) detector with a noise equivalent power (NEP) of 0.1 pW/Hz$^{1/2}$, the optimum online optical depth for minimum noise is around 0.7 (0.9) for 3.9 (4.5) µm. Due to the lack of more comprehensive mid-IR detector data the NEP is estimated on the basis of near-IR realizations and mid-IR prototypes (Sun et al., 2017; Martyniuk et al., 2023) using large security margins, likely representing a worst case. The absence of strong lines in the 3.9 µm band (Fig. 1) prevents an optimum online setting yet the low offline

optical depth of 0.03 provides satisfying low-noise levels (Fig. 5). At 4.5 µm, a trough nearby the optimum optical depth could be selected (Fig. 2). Stronger neighboring troughs that would fit the optimum suffer from overlapping water line wings and larger temperature sensitivity. The 4.5 µm selection also allows for a measurement precision fulfilling the initial requirement of 0.5 %, although at a higher noise level, primarily due to the relatively high offline optical depth of 0.33. Away from the optimum, noise increases towards lower optical depths because of a smaller DAOD (Eq. 3), and towards

higher optical depths because of online signal attenuation.

In addition to uncorrelated random noise which averages out over longer data accumulation lengths, persisting systematic uncertainties (biases) may arise from errors in the spectroscopic parameters. The HITRAN database specifies the $N_2O$ line intensity uncertainties to within 2 and 5 %. This does not threaten the objectives to quantify agricultural areal or point source emissions since those are derived from relative column gradients between sources and background rather than from absolute

measurements. In addition, spectroscopy errors can be corrected by comparing the lidar columns against collocated profiles from high accuracy aircraft in-situ sensors (Amediek et al., 2017; Mao et al., 2024). Biases may also arise from variability in the actual aerosol, temperature and pressure profiles within the columns. Experience and simulation (Ehret et al., 2008; Kiemle et al., 2011; Amediek et al., 2017) shows that these usually remain below 1 %. Finally, albedo variations cause measurement uncertainties if the on- and offline surface laser spots are not fully overlapping, which is generally the case.

Those however tend to behave like random deviations leading to slightly increased noise levels (Amediek et al., 2009).

## 5 Technology Options and Readiness

IR lidar transmitters that can be considered for the described purpose are (a) tunable solid-state lasers such as transition-metal (TM)-doped II-VI chalcogenide lasers, for example Fe:ZnS or Fe:ZnSe lasers, (b) optical parametric oscillators (OPO) and amplifiers, and (c) laser sources based on nonlinear difference frequency generation (DFG) or Raman shifting.

Comprehensive overviews on recent advances in those laser-based mid-infrared sources are given in Vodopyanov (2020) and Ren et al. (2021). Using OPOs, wavelengths in the 3.9 to 4.5-µm range are readily accessible. When pumping with the ubiquitous Nd:YAG laser at 1.064 µm, the corresponding signal wavelengths that lead to an idler wavelength at 3.9 or 4.5 µm are ~1.46 or 1.39 µm, respectively. This requires suitable nonlinear crystals with transparency at all these wavelengths. But, other pump laser sources at longer wavelength such as 2.05 µm (e.g. Medina et al., 2021), cascaded OPOs or DFG

sources are also options (e.g. DFG of 1.064 μm with 1.46 μm results in 3.9 μm and mixing 1.064 μm with 1.39 μm results in 4.5 μm, respectively). Using DFG, lidar measurements of species such as hydrocarbons in the wavelength range around 3.4 μm have been performed (Robinson et al., 2011; Gardiner et al., 2017).

Low-noise and high-bandwidth radiation detection is a challenge in the mid-IR. Prototype MCT APDs can achieve very low excess noise in the mid-IR, yet cooling down to at least 200 K is required to reduce dark currents. Sun et al. (2017 and 2021) report a detector linear analog output with a dynamic range of 2-3 orders of magnitude at a fixed APD gain for MCT material that has a cut-off at 4.3 μm. These detectors work well at 3.9 μm yet have little response at 4.5 μm. Generally, developments are ongoing but manufacturability is considered low (Chen et al., 2021; Martyniuk et al., 2023). In addition, the literature reveals that efforts go into the development of imaging sensors, while lidar requires a single sensor with a large photosensitive area in order to satisfy optical constraints (cf. Table 3). Due to the lack of data we used in our study a conservative approach for the simulation, with a large security margin for the detector noise on the basis of data from near-IR realizations and mid-IR prototypes. An alternative to MCT is indium antimonide (InSb), yet mid-IR InSb APDs apparently have more noise and less bandwidth (Abautret et al., 2015; Alimi et al., 2020). Besides APD detectors, up-conversion detectors (UCD; Hoegstedt et al., 2016; Meng et al., 2018) or superconducting nanowire single-photon detectors (SNSPD) could offer high-efficiency, low-noise signal detection, yet SNSPDs require operating temperatures below 3 K which is challenging onboard aircraft. A single-photon response was reported up to 25 K at 1.5 μm (Charaev et al., 2023), unfortunately not much less challenging. Also, in this area, developments are underway.

Concerning methodical alternatives for active remote sensing, the more common differential absorption lidar (DIAL) technique exploits atmospheric instead of surface backscatter with the advantage of providing vertical profiles of trace gases instead of column concentrations. A dedicated simulation using the same lidar simulator shows that this is at the expense of a 100 times larger laser power and telescope aperture product ($P_L \cdot A$ = 0.3 W m$^2$ instead of 0.003 W m$^2$ from Table 3) to compensate for the roughly 100 times weaker atmospheric backscatter signals (in terms of laser photons per range gate), even within a boundary layer with rural aerosol. A low-power alternative is IPDA or DIAL with heterodyne instead of direct signal detection. It requires a diffraction-limited optical setup and laser pulse repetition rates in the kHz range to manage speckle-induced noise. So far, ground-based profiling systems (Koch et al., 2008; Yu et al., 2024) and an airborne realization (Spiers et al., 2011; Jacob et al., 2019) for $CO_2$ have been reported, yet only in the near-IR. Another low-power option for IPDA is (modulated) continuous-wave (cw) laser operation instead of emitting short pulsed signals (e.g., Campbell et al., 2020). For measurements with a precision requirement below 1 % however, the length of the atmospheric column must be known to an accuracy of better than 3 m which is only practicable with short laser pulses in combination with a sufficiently large detection bandwidth (Table 3; Ehret et al., 2008). Alternatively, a precision range finder had to be added which annihilates the cost benefit of cw lidar. We conclude that methodical alternatives for $N_2O$ active remote sensing are either too expensive, or their maturity for airborne operation lags behind that of direct detection pulsed IPDA.

## 6 Conclusion

With a resulting noise level of < 0.5 % an airborne IPDA lidar provides important new capabilities for $N_2O$ regional gradient or hot spot detection with technically realizable and affordable transmitter (100 mW average laser power) and receiver (20 cm telescope, MCT APD) characteristics. Using an MCT APD detector that requires cooling to 200 K the system could fit into a small- to mid-size research aircraft. The simulation results show better performance at 3.9 µm in terms of the noise level. On the other hand, the trough position at 4.5 µm yields higher measurement sensitivity at low altitudes and considerably relaxes the laser frequency stability requirement. Which option is finally preferred will depend on many factors including several aspects of laser technology such as availability, complexity, linewidth, frequency locking performance, as well as detector availability, costs, aircraft suitability, etc. The simulation tool will be applied to trade off various instrument options as technology is developing. Better low-noise IR detectors will be particularly beneficial. While a satellite implementation is not impossible, but still far away because of lacking space-proof technology, the development of an airborne $N_2O$ lidar at 3.9 or 4.5 µm is recommended given the technical feasibility and the scientific-societal need.

## Code/Data availability

Code and data can be provided by the first author on request.

## Author contribution

CK developed the model code, performed the simulations, and prepared the manuscript with contributions from all co-authors. AF, CF, GE and MW contributed information on available laser and detector technology.

## Competing interests

The authors declare that they have no conflict of interest. CK is member of the editorial board of AMT.

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
