# Peer review of "Design Study for an Airborne N2O Lidar"

_EGUsphere, 2024_

## Referee Comment (RC1)

Dear Editor,                                                                                       August 11, 2024

I have reviewed the manuscript below, which was submitted to the journal *Atmospheric Measurement Techniques:*

"Design Study for an Airborne N2O Lidar" by authors Christoph Kiemle, Andreas Fix, Christian Fruck, Gerhard Ehret, and Martin Wirth.

General comments:

The manuscript describes the scientific need for an airborne IPDA lidar to measure N2O the atmospheric column below the aircraft. The manuscript infers some initial measurement requirements and shows the results of an initial study to assess suitable spectral regions to measure N2O. The authors assess 3 possible spectral regions for the measurement. They use a lidar model to estimate the random error of measurements from notional designs that operate in the 3.9 and 4.5 um spectral regions. It also gives an overview of possible lasers and detectors for such a lidar and recommends some next steps to improve their suitability for an airborne IPDA lidar.

Recommendation:

This manuscript addresses designing a new type of direct detection IPDA lidar to address an important unmet need for monitoring the atmospheric concentrations of N2O, the 3$^{rd}$ most important greenhouse gas. It clearly shows the importance of measuring N2O, the challenges of previous techniques and the significant benefits of using a lidar to fill this need. The manuscript also includes many relevant references.

The manuscript covers an important new topic and is a good match to the emphasis of this journal. I found that the basics of the manuscript's concept study appear sound. Nonetheless, I did find a number of areas where the manuscript would benefit from revisions or minor improvements. I therefore recommend accepting the manuscript, but only after recommendations and comments below have been addressed.

Mandatory changes:

1.  Page 1 line 13. "best option". The manuscript actually shows that 4.5 um is best only in the sense of requiring less laser frequency stability, while the 3.9 um version has lower measurement error. (See more in comment 22 below). Please change wording to reflect this.

2.  Page 1 line 10 The term "terrestrial radiation" is used several places in the manuscript. Do the authors mean thermal emission from the surface? If so then "thermal emission" seems better terminology.

3.  Page 2 line 42. Please explain why measurements at dawn dusk or at night may be important.

4.  Page 2 line 48. Please add that the strength of signals measured by IPDA lidar is orders of magnitude stronger than atmospheric backscatter signal measured by DIAL.

5.  Page 2 line 51. Please add the reference X.Sun et al. AMT, 2021 that shows high measurement accuracy of an airborne Co2 lidar (in comparison to in situ) in its Figures 7 & 8.

6.  Page 2 measurement requirement discussion. Lidar measurement stability (ie lack of drift or changing bias) is very important, particularly for airborne campaigns that fly over different surfaces and atmospheric conditions for hours. Please add an estimate of how stable the airborne lidar measurements need to be to be useful in the example airborne N2O sample shown in Eckl 2021, Figure 1b.

7.  Page 3, line 75 OD equation. Please add lamdas (wavelength term) to the OD and sigma terms.

8.  Page 3 Line 78. Please add gas number density to the line selection criteria.

9.  Figures 1-3. The colors of N2O & CH4 in the plots are similar and hard to differentiate. Please change the color of one of them to clarify.

10. Figures 1& 2. Please add a figure showing an expanded scale of spectral regions & lines in each band recommended for the on & offline wavelengths.

11. Page 5 upper paragraph discussion on-line candidates. Please discuss the laser linewidth requirements for each band and add those to the specification summary in Table 3.

12. Page 6 line 152. Discussion of surface albedo. Since the reflection from water surfaces is specular reflectors (not diffuse like that from land) the strength of their reflected laser signal back to the lidar receiver can vary over orders of magnitude since it depends on factors like surface roughness and off-nadir angle. Since this manuscript primarily addresses measurements over land, I suggest just omitting the mention of water surfaces here.

13. Page 7 Table 2. The thermal emission from the surface is a strong function of temperature. Please give the assumed surface temperature for the calculated emission values.

14. Page 8 Equation 3. For IPDA lidar DAOD is computed from the energies of the transmitted and reference pulses (E), not their optical power (P). This is because the optical power of the pulses varies with time during the pulse, and the peak optical power from the reflected pulse also depends on the range spreading caused by topographic roughness. Please correct the equation.

15. Page 8 Table 8. Please add the assumed values for laser divergence and laser linewidths.

16. Page 9 SNR discussion. Speckle noise can be more significant for laser measurements in the mid-IR since the speckle cell sizes are larger and there are fewer "speckles" (regions of constructive interference) on the detector surface. Please add a discussion about the errors from speckle noise.

17. Page 9 Figure 5. Please explain the cause of the error floors (asymptotes) for the ideal noiseless detector plots shown in Figure 5.

18. Page 11. Detector availability is an important factor when considering which spectral band to use. The results on several MCT detectors reported by by X. Sun et al. in references 2017 & subsequent years all use MCT material that has a cutoff at 4.3 um. See Figure 2. Although these detectors work well at 3.9 um but they have little response at 4.5 um. This should be mentioned.

19. Page 10 laser discussion. Please provide at least one reference for each of the candidate laser approaches mentioned.

20. Page 11. Line 256. Please mention the operating temperature required for superconducting nanowire detectors.

21. Page 11 Line 268. Range precision & short pulses. Amediek et al 2013 showed that m-level range precision was obtained with the CO2 Sounder lidar that transmitted 1-usec wide rectangular laser pulses. Please update the range precision statement.

22. Page 12 line 278. Which option is preferred depends on many factors including several aspects of laser technology (including availability, complexity, linewidth, frequency locking performance), detector availability, costs, etc. Please revise the preference statement to better reflect the many aspects of the wavelength choice.

Recommended minor edits & wording changes:

1.  Page 1 line 8 – the study addresses a lidar to measure atmospheric concentrations, not emissions (which also needs wind speeds). Please revise.

2.  Page 1 line 18 Recommend changing "space-proof" to "space"

3. Page 1 line 18. Suggest rewording last sentence to mention that demonstrating airborne lidar has been almost always viewed as a required precursor to space versions.

4. Page 1 line 26. "source" -> sources.

5. Page 2 line 41 "will" -> has the potential to

6. Page 3 line 65. "ideas" -> concepts.

7. Page 6 line 132. "characteristics of both band selections" -> candidate lines in both bands.

8. Figure 4 Please move the labels for "center" and "trough" away from the plot frame.

9. Page 10, line 227. Change "saturation" to attenuation.

10. Page 11 line 249. Detector dynamic range. A linear dynamic range of 2-3 orders of magnitude is quite good. Please delete the word only".

11. Page 12 line 274. Please restating "design fulfills the measurement requirements" as "provides important new measurement capabilities" or something similar.

---

## Author Comment (AC1)

Dear Editor,

our answers to all comments of reviewer #2 are embedded in red.

The manuscript focuses on the specifications of an N2O lidar system suitable for agricultural emission concentrations. It is an important challenge, and the authors make a good description of a proposed system with relatively high technological readiness, though implementation is still a challenge. The general ideas and concepts seem reasonable and a good contribution to this relatively unexplored issue.

As this is a technological modeling investigation, I have relatively few comments.

Comments:

Terrestrial radiation is used throughout, but it's unclear – I assume they mean thermal? That would be more precise.

We replaced "terrestrial radiation" by "thermal emission" throughout the manuscript.

Line 152: Is water possible, given the reflections are not diffused in the same was as terrestrial sources? If this is applicable to water (e.g., wastewater ponds, oceans) then that should be better supported.

Lidar measurements over water are possible. However, since the manuscript primarily addresses measurements over land, we omitted mentioning water surfaces, as reviewer 1 suggested.

Table 2: What are the thermal conditions assumed for this table?

We assumed the "worst case" thermal emission of a blackbody with 288 K (the earth surface average temperature) and no atmosphere. We added this information in the text and the table.

Line 261: Are there safety concerns at this wavelength/power that may preclude use in agriculture?

No. In the wavelength range > 2.6 µm the human eye is not transparent for radiation and, thus, the exposure limit for the human eye and skin are identical, and uncritical.

Line 267: Please provide a citation for the modulated continuous wave approach.

Campbell, J. F., Lin, B., Dobler, J., Pal, S., Davis, K., Obland, M. D., et al.: Field evaluation of column CO2 retrievals from intensity-modulated continuous-wave differential absorption lidar measurements during the ACT-America campaign. Earth and Space Science, 7, e2019EA000847. https://doi.org/10.1029/2019EA000847, 2020. Included in the manuscript.

Discussion: A table outlining the technological options, strengths, and weaknesses would be helpful.

Given the actual technological progress and unavailable details on many components we think a table may be anticipating current progress too early with the risk to bias or mislead opinions.

Smaller comment:

Is the instrumentation described feasible to fit into an airplane that can fly that those altitudes and speeds?  This is especially relevant for the instrumentation options that require active cooling.

Using detectors (MCT APD) that require cooling to 200 K the system could fit into a small- to mid-size research aircraft. We added this to the conclusion.

---

## Author Comment (AC2)

Dear Editor,

our answers to all comments of reviewer #1 are embedded in red.

General comments:

The manuscript describes the scientific need for an airborne IPDA lidar to measure N2O the atmospheric column below the aircraft. The manuscript infers some initial measurement requirements and shows the results of an initial study to assess suitable spectral regions to measure N2O. The authors assess 3 possible spectral regions for the measurement. They use a lidar model to estimate the random error of measurements from notional designs that operate in the 3.9 and 4.5 um spectral regions. It also gives an overview of possible lasers and detectors for such a lidar and recommends some next steps to improve their suitability for an airborne IPDA lidar.

Recommendation:

This manuscript addresses designing a new type of direct detection IPDA lidar to address an important unmet need for monitoring the atmospheric concentrations of N2O, the 3rd most important greenhouse gas. It clearly shows the importance of measuring N2O, the challenges of previous techniques and the significant benefits of using a lidar to fill this need. The manuscript also includes many relevant references.

The manuscript covers an important new topic and is a good match to the emphasis of this journal. I found that the basics of the manuscript's concept study appear sound. Nonetheless, I did find a number of areas where the manuscript would benefit from revisions or minor improvements. I therefore recommend accepting the manuscript, but only after recommendations and comments below have been addressed.

Mandatory changes:

1. Page 1 line 13. "best option". The manuscript actually shows that 4.5 um is best only in the sense of requiring less laser frequency stability, while the 3.9 um version has lower measurement error. (See more in comment 22 below). Please change wording to reflect this.

   We replaced "best" to "favored"

2. Page 1 line 10 The term "terrestrial radiation" is used several places in the manuscript. Do the authors mean thermal emission from the surface? If so then "thermal emission" seems better terminology.

   We replaced "terrestrial radiation" by "thermal emission" throughout the manuscript

3. Page 2 line 42. Please explain why measurements at dawn dusk or at night may be important.

   They may be important to study the diurnal concentration evolution. We added this to the text.

4. Page 2 line 48. Please add that the strength of signals measured by IPDA lidar is orders of magnitude stronger than atmospheric backscatter signal measured by DIAL.

   done

5. Page 2 line 51. Please add the reference X.Sun et al. AMT, 2021 that shows high measurement accuracy of an airborne Co2 lidar (in comparison to in situ) in its Figures 7 & 8.

   done

6. Page 2 measurement requirement discussion. Lidar measurement stability (ie lack of drift or changing bias) is very important, particularly for airborne campaigns that fly over different surfaces and atmospheric conditions for hours. Please add an estimate of how stable the airborne lidar measurements need to be to be useful in the example airborne N2O sample shown in Eckl 2021, Figure 1b.

   We added: "…, the measurement instability due to instrumental drifts or changing biases should remain below 0.5 %. Experience from airborne lidar campaigns shows that long-term stability can be controlled by executing repeated flight legs over the same tracks, and over background concentrations in case those can be assumed constant."

7. Page 3, line 75 OD equation. Please add lamdas (wavelength term) to the OD and sigma terms.

done

8. Page 3 Line 78. Please add gas number density to the line selection criteria.

   done

9. Figures 1-3. The colors of N2O & CH4 in the plots are similar and hard to differentiate. Please change the color of one of them to clarify.

   Done

10. Figures 1& 2. Please add a figure showing an expanded scale of spectral regions & lines in each band recommended for the on & offline wavelengths.

    Done

11. Page 5 upper paragraph discussion on-line candidates. Please discuss the laser linewidth requirements for each band and add those to the specification summary in Table 3.

    Fig. 10 in Ehret et al. 2008 shows that the laser linewidth (bandwidth) requirements are relatively uncritical. However, we refer to Fig. 11 in Ehret et al. 2008 to discuss the laser frequency uncertainty (jitter) issues: for a laser with a (spectral) bandwidth of 30 MHz, a laser frequency uncertainty of 1.2 MHz leads to a bias of ~0.35 % in the vicinity of a CO2 line center comparable to the N2O line selected at 3.9 μm. For a CH4 trough position roughly comparable to the one selected in the 4.5 μm N2O band, Kiemle et al., 2011 state in their section 3 that the frequency uncertainty requirement can be relaxed by a factor of fifty to hundred, compared to a line center or wing position. We have added appropriate text to this paragraph and to Table 3.

12. Page 6 line 152. Discussion of surface albedo. Since the reflection from water surfaces is specular reflectors (not diffuse like that from land) the strength of their reflected laser signal back to the lidar receiver can vary over orders of magnitude since it depends on factors like surface roughness and off-nadir angle. Since this manuscript primarily addresses measurements over land, I suggest just omitting the mention of water surfaces here.

    omitted

13. Page 7 Table 2. The thermal emission from the surface is a strong function of temperature. Please give the assumed surface temperature for the calculated emission values.

    We assumed a temperature of 288 K (the earth surface average). We added this information in the text and the table.

14. Page 8 Equation 3. For IPDA lidar DAOD is computed from the energies of the transmitted and reference pulses (E), not their optical power (P). This is because the optical power of the pulses varies with time during the pulse, and the peak optical power from the reflected pulse also depends on the range spreading caused by topographic roughness. Please correct the equation.

    corrected

15. Page 8 Table 8. Please add the assumed values for laser divergence and laser linewidths.

    done

16. Page 9 SNR discussion. Speckle noise can be more significant for laser measurements in the mid- IR since the speckle cell sizes are larger and there are fewer "speckles" (regions of constructive interference) on the detector surface. Please add a discussion about the errors from speckle noise.

    Speckle noise is considered in the performance simulations. The effect can be compensated by a larger telescope field-of-view and laser beam divergence, compared to near-IR applications. A mention was added.

17. Page 9 Figure 5. Please explain the cause of the error floors (asymptotes) for the ideal noiseless detector plots shown in Figure 5.

    There is no asymptotic behavior for the ideal detector. Just the detector noise is assumed zero, yet all other noise contributors remain: Poisson noise from laser photons and background radiation, laser speckle noise within the field of view, and energy reference measurement noise. The curves are flatter since a low-noise instrument is more tolerant

with respect to suboptimal spectroscopic settings, and their minima are right-shifted because the optimum online optical depth is larger under low-noise conditions (Ehret et al., 2008). However, towards higher than optimum optical depths (not shown here) the noise increases again, also for the ideal detector because of online signal attenuation/saturation. We added a sentence to make this clearer in the text.

18. Page 11. Detector availability is an important factor when considering which spectral band to use. The results on several MCT detectors reported by by X. Sun et al. in references 2017 & subsequent years all use MCT material that has a cutoff at 4.3 um. See Figure 2. Although these detectors work well at 3.9 um but they have little response at 4.5 um. This should be mentioned.

    done

19. Page 10 laser discussion. Please provide at least one reference for each of the candidate laser approaches mentioned.

    Done. We found and cite comprehensive overviews on recent advances in laser-based mid-infrared sources in Vodopyanov (2020) and Ren et al. (2021).

20. Page 11. Line 256. Please mention the operating temperature required for superconducting nanowire detectors.

    Actual SNSPDs require operating temperatures below 3 K which is challenging onboard aircraft, yet a single-photon response has already been reported up to 25 K, at least at 1.5 µm (Charaev et al., 2023). We added this in the text.

21. Page 11 Line 268. Range precision & short pulses. Amediek et al 2013 showed that m-level range precision was obtained with the CO2 Sounder lidar that transmitted 1-usec wide rectangular laser pulses. Please update the range precision statement.

    done

22. Page 12 line 278. Which option is preferred depends on many factors including several aspects of laser technology (including availability, complexity, linewidth, frequency locking performance), detector availability, costs, etc. Please revise the preference statement to better reflect the many aspects of the wavelength choice.

    done

Recommended minor edits & wording changes:

1. Page 1 line 8 – the study addresses a lidar to measure atmospheric concentrations, not emissions (which also needs wind speeds). Please revise.

    done

2. Page 1 line 18 Recommend changing "space-proof" to "space"

    done

3. Page 1 line 18. Suggest rewording last sentence to mention that demonstrating airborne lidar has been almost always viewed as a required precursor to space versions.

   done

4. Page 1 line 26. "source" -> sources.

   done

5. Page 2 line 41 "will" -> has the potential to

   done

6. Page 3 line 65. "ideas" -> concepts.

   done

7. Page 6 line 132. "characteristics of both band selections" -> candidate lines in both bands.

   done

8. Figure 4 Please move the labels for "center" and "trough" away from the plot frame.

   done

9. Page 10, line 227. Change "saturation" to attenuation.

   done

10. Page 11 line 249. Detector dynamic range. A linear dynamic range of 2-3 orders of magnitude is quite good. Please delete the word only".

    done

11. Page 12 line 274. Please restating "design fulfills the measurement requirements" as "provides important new measurement capabilities" or something similar.

    done